# Synthesis and Characterization of Lithium-Ion Conductive LATP-LaPO_4_ Composites Using La_2_O_3_ Nano-Powder

**DOI:** 10.3390/ma14133502

**Published:** 2021-06-23

**Authors:** Fangzhou Song, Masayoshi Uematsu, Takeshi Yabutsuka, Takeshi Yao, Shigeomi Takai

**Affiliations:** 1Graduate School of Energy Science, Kyoto University, Yoshida-Honmachi, Sakyo-Ku, Kyoto 606-8501, Japan; song.fangzhou.48a@st.kyoto-u.ac.jp (F.S.); kagawa18@gmail.com (M.U.); yabutsuka@energy.kyoto-u.ac.jp (T.Y.); 2Kyoto University, Yoshida-Honmachi, Sakyo-Ku, Kyoto 606-8501, Japan; t_yao@hera.eonet.ne.jp

**Keywords:** insulative particle dispersion, lithium-ion conductor, LATP, all-solid-state battery

## Abstract

LATP-based composite electrolytes were prepared by sintering the mixtures of LATP precursor and La_2_O_3_ nano-powder. Powder X-ray diffraction and scanning electron microscopy suggest that La_2_O_3_ can react with LATP during sintering to form fine LaPO_4_ particles that are dispersed in the LATP matrix. The room temperature conductivity initially increases with La_2_O_3_ nano-powder addition showing the maximum of 0.69 mS∙cm^−1^ at 6 wt.%, above which, conductivity decreases with the introduction of La_2_O_3_. The activation energy of conductivity is not largely varied with the La_2_O_3_ content, suggesting that the conduction mechanism is essentially preserved despite LaPO_4_ dispersion. In comparison with the previously reported LATP-LLTO system, although some unidentified impurity slightly reduces the conductivity maximum, the fine dispersion of LaPO_4_ particles can be achieved in the LATP–La_2_O_3_ system.

## 1. Introduction

The popularization of electric vehicles and mobile devices is calling for an advance in battery technology to meet the requirement on the battery reliability and higher energy density. Solid-state electrolytes (SSEs), with wider electrochemical window, nonflammability and low-temperature stability in comparison with the liquid counterparts, is a key component for the all-solid-state battery (ASSB) that is safer to use and allows more compact designs [1,2,3,4]. In recent decades, research has focused on the improvement of room temperature conductivities for SSEs, mainly through the development of new lithium-ion conductors or the improvement of currently available SSEs by means of doping or lattice tuning [1,3,5,6,7,8,9].

In addition to the above strategies, insulator particle dispersion has been explored to improve lithium-ion conduction, which was originally reported in C. Liang’s work where Al_2_O_3_ particles were dispersed in LiI and resulted in 50 times enhancement in conductivity [10]. Similar phenomena were observed in subsequent research on halide-type lithium-ion conductors [11,12,13,14,15,16]. To explain the mechanism, various types of theories have been developed [17], among which, a space charge layer model originated by C. Wagner [18] is mostly accepted. According to this model, the charge carriers at the interface between ion conductor and insulative particles are redistributed due to the difference in chemical potentials, leading to deviation from electroneutrality to form the favorable region for the charge carrier to migrate [6,19,20,21,22,23,24,25]. Recent studies by means of NMR characterization also support the space charge layer model [26,27]. This strategy has been applied to a limited range of lithium-ion conductors such as halides and LiBH_4_ [10,11,12,13,14,15,20,21,22,23,24,25,27,28,29,30,31,32,33,34]. Although it was recently reported that conductivity for oxide-based lithium-ion conductors can be increased by adding a secondary phase to modify the grain boundary conductivity [35,36,37], relatively few studies have focused on the insulator particle dispersion strategy in oxide-based materials [38,39,40,41].

Li_1.3_Al_0.3_Ti_1.7_(PO_4_)_3_ (LATP) is an oxide-based solid-state electrolyte with a rhombohedral NASICON-type structure that is composed of corner-sharing MO_6_ (M = Ti or Al) octahedra and PO_4_ tetrahedra, forming a three-dimensional diffusion network for lithium-ions within the lattice [1,3]. We have previously achieved 3 times improvement in room temperature conductivity by introducing Li_0.348_La_0.55_TiO_3_ (LLTO) particles into the LATP matrix. The introduced LLTO reacted with the LATP matrix during the sintering process, forming fine LaPO_4_ which act as insulative particles [40]. However, the direct introduction of LaPO_4_ into LATP did not enhance the conductivity due to the growth of LaPO_4_ particles [42]. In order to disperse the LaPO_4_ particles finely through a simplified reaction, La_2_O_3_ nano-powder is selected as a more direct lanthanum source rather than LLTO particles. In this work, LATP–LaPO_4_ composites are prepared by employing La_2_O_3_ nano-powder to compare with the results of the previous LLTO added system.

## 2. Materials and Methods

### 2.1. Synthesis of the LATP Precursor

Li_1.3_Al_0.3_Ti_1.7_(PO_4_)_3_ (LATP) precursor was prepared by the solid-state reaction method. Stoichiometric amounts of Li_2_CO_3_ (99.0% Wako Pure Chem., Osaka, Japan, with 10 wt.% excess), γ-Al_2_O_3_ (97.0% Stream Chemical, Newburyport, MA, USA), TiO_2_ (rutile, 99.9% High Purity Chem., Saitama, Japan) and NH_4_H_2_PO_4_ (99.0% Wako Pure Chem., Osaka, Japan) were mixed in an automatic grinder for 5 h with an aid of ethanol. After drying for 24 h, the mixture was uniaxially pressed to form the green compact which was then calcined at 700 °C for 2 h. To form fine LATP precursor, the calcined product was crushed and ball-milled in zirconia pot with ethanol and zirconia balls for 5 h at 400 RPM (Pulverisette7 Premium Line, Fritsch, Idar-Oberstein, Germany).

### 2.2. Synthesis of the LATP–La_2_O_3_ Composite

To fabricate LATP-La_2_O_3_ composite pellets, the fine LATP precursor was mixed with La_2_O_3_ nano-powder (<100 nm, 99% Sigma-Aldrich, Hesse, Germany) by ball milling (zirconia balls and pot, Pulverisette7 Premium Line, Fritsch) with the aid of a small amount of ethanol for 1.5 h at 400 RPM. After drying, the powder mixture was isostatically pressed to form cylindrical pellets at 200 MPa followed by sintering at 1000 °C for 4 h. The sintering time was optimized according to the preliminarily examined sintering time dependence, as represented in Appendix A in the Appendix A. In this work, the introduced La_2_O_3_ nano-powders were weighted 2, 4, 6, 8, 12 and 16 wt.% of the total weight (LATP + La_2_O_3_ mixture). Herein, the samples are referred as LATP–*x* wt.% La_2_O_3_, based on the amount of added La_2_O_3_.

### 2.3. Characterizations and Electrochemical Properties

The obtained crystalline phases were investigated by powder XRD on the Ultima VI diffractometer (Rigaku, Tokyo, Japan) using a CuKα radiation source (40 kV, 40 mA). The microstructure and particle distribution of the samples were observed by scanning electron microscopy under the back-scattering electron mode (SEM, SU6600, Hitachi, Tokyo, Japan). The sample pellets with a 6 mm diameter and 3 mm thickness were polished on both sides and sputtered with gold to form electrodes. To investigate the temperature variation of electrochemical impedance, the samples were clamped in a 4-electrode test apparatus in a temperature-controlled tubular furnace. An amount of 0.5 V of AC potential was applied to the sample pellets using an LRC meter (3531 Z Hitester, Hioki, Japan) in a frequency range of 130 Hz–1.3 MHz and a temperature range of 25–200 °C. The conductivities were calculated by the equivalent circuit fitting from the impedance spectroscopies using ZView^®^ software (Scribner, New York, NA, USA) [43].

## 3. Results and Discussions

Powder XRD pattern of LATP (Li_1.3_Al_0.3_Ti_1.7_(PO_4_)_3_) and LATP–*x* wt.% La_2_O_3_ composites are shown in Figure 1, where the major peaks are associated with LATP that is isostructural with LiTi_2_(PO_4_)_3_. The existence of LaPO_4_ (labelled by solid inverted triangle) suggests a solid-state reaction between the LATP matrix and introduced La_2_O_3_ during sintering. LaPO_4_ formation at the sintering also occurred in LATP-LLTO and LAGP-LLTO systems in the previous works [40,41]. In addition to LaPO_4_ formation, a LiTiPO_5_ phase and an unidentified impurity were also observed in the powder XRD patterns, as labelled by hollow diamonds and hollow inverted triangles in Figure 1. The small amount of LiTiPO_5_ phase is believed to be formed during sintering when the LATP matrix donates phosphorus to form LaPO_4_. The LiTiPO_5_ and unidentified impurities constantly remained despite prolonged sintering, as observed in Appendix A, for the LATP–8 wt.% La_2_O_3_ system.

Figure 2 presents SEM images of pristine LATP and composite samples captured under back-scattered electron mode, where the bright spots represent the lanthanum-containing particles due to the heaver atom. For relatively smaller La_2_O_3_ addition below 8 wt.%, the dispersed particles are isolated, keeping the similar sizes, as shown in Figure 2b–d. At higher La_2_O_3_ additions such as 12 or 16 wt.%, the particles are aggregated to break the percolation of LATP matrix, as shown in Figure 2e,f.

The Nyquist plots of electrochemical impedance spectroscopies for pristine LATP and composite samples are shown in Figure 3. Owing to the limited frequency range, the impedance spectra are fitted by using a conventional equivalent circuit in the inset to obtain the right side of the semi-circles as the total resistivity. The room temperature conductivities of the samples are presented as a function of La_2_O_3_ addition in Figure 4, where the highest conductivity of 0.69 mS∙cm^−1^ is achieved at 6 wt.% of La_2_O_3_ addition. This suggests that the addition of La_2_O_3_ nano-powder can form LaPO_4_ particles in LATP matrix. From 6 wt.% up to 16 wt.% of La_2_O_3_ introduction, the conductivity decreases with the La_2_O_3_ addition. This is caused by the aggregation of the insulative particles, which severely block the migration of the lithium-ions in the LATP matrix to reduce the total conductivity.

For comparison, the conductivity of previous LATP–LLTO composites [40] are also plotted in Figure 4 (hollow triangles). The weight percentage of LLTO is converted to the equivalent amount of La_2_O_3_ based on the lanthanum content in additives. Although the highest conductivity in this work is slightly smaller than the previously observed 0.76 mS∙cm^−1^ in LATP–4 wt.% LLTO [40], about three-fold enhancement from the pristine can be achieved. The slightly smaller conductivity might be due to the unidentified impurity, which could block the LATP matrix/LaPO_4_ particle interface. It should be noted that the maximum conductivity occurs at higher lanthanum content in comparison with the previous LATP–LLTO system, indicating that La_2_O_3_ nano-powder is effective in forming finely dispersed LaPO_4_ particles without aggregation. Suppressing the formation of unidentified impurity should be critical for further enhancement in conductivity.

The conductivities are plotted against inverse temperature, as shown in Figure 5a, which can be linearly fitted to the Arrhenius equation *σ*_T_*T* = *σ*_0_ exp(−*E*_a_/*kT*), where *σ*_T_, *σ*_0_ and *E*_a_ denote the total conductivity, pre-exponential term and the activation energy, respectively. The deduced activation energy is plotted as a function of La_2_O_3_ addition in Figure 5b. The activation energies are similar to pristine LATP or slightly increased with the introduction of La_2_O_3_ nano-powder, suggesting that the lithium migration mechanism of composite is essentially consistent with that of pristine LATP.

In summary, by adding La_2_O_3_ nano-powder into the LATP precursor, LaPO_4_ particles can be dispersed into the LATP matrix through solid-state reaction during sintering process. A three-fold enhancement in conductivity is observed in the LATP–6 wt.% La_2_O_3_ sample, while the activation energy of the composite is not largely different from the pristine LATP. In further study, characterizations such as ^7^Li solid-state NMR spectroscopy and high-resolution transmission electron microscopy are required to scrutinize the lithium-ion conduction mechanism and microstructural features at the LATP matrix/LaPO_4_ particle interface.

## 4. Conclusions

In this work, LATP-based composite electrolytes were synthesized by adding La_2_O_3_ nano-powder into an LATP precursor. Powder XRD and back-scattered SEM prove that LaPO_4_ particles were formed to disperse in the sintered samples during sintering. The aggregation of particles is observed at higher lanthanum introduction. The room temperature conductivity of the composite electrolytes increases with the La_2_O_3_ addition until 6 wt.%, where the maximum conductivity of 0.69 mS∙cm^−1^ is achieved, which is ascribed to the insulative particle dispersion effect. In comparison with the previous study on the LATP–LLTO composites [40], the maximum conductivity is observed at the higher lanthanum content, although the maximum conductivity is inferior to the previous one. Further improvement is expected through the elimination of impurities. The compositional dependence of activation energies of conductivity suggests that the present LATP-La_2_O_3_ system possesses a similar conduction mechanism to the previous LATP-LLTO system.

## Figures and Tables

**Figure 1 materials-14-03502-f001:**
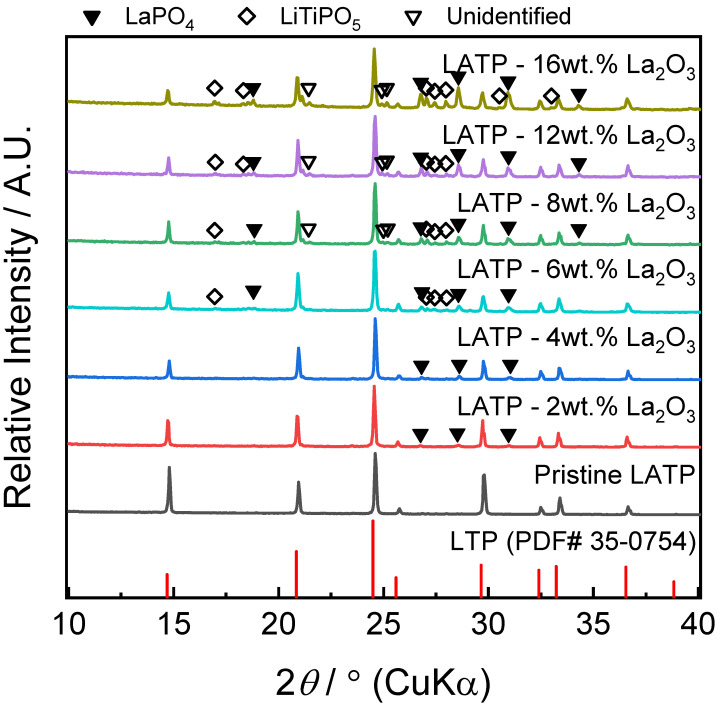
Powder XRD patterns of pristine LATP and LATP–La_2_O_3_ composites. LaPO_4_, LiTiPO_5_ and unidentified phases are labelled by solid inverted triangle, hollow diamond, and hollow inverted triangle, respectively.

**Figure 2 materials-14-03502-f002:**
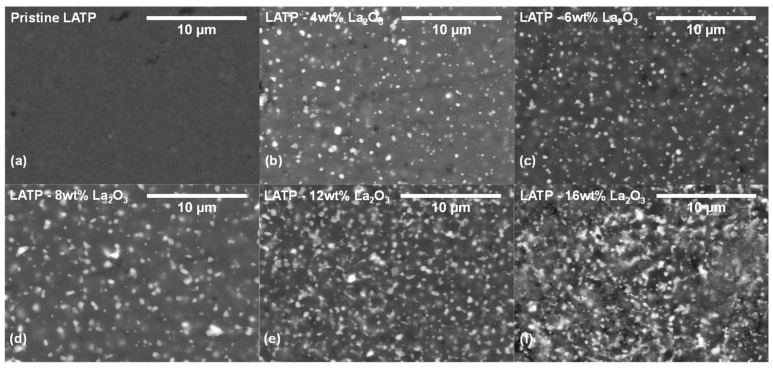
Back-scattering SEM images of (**a**) pristine LATP, (**b**) LATP–4 wt.%, (**c**) LATP–6 wt.%, (**d**) LATP–8 wt.%, (**e**) LATP–12 wt.%, and (**f**) LATP–16 wt.% La_2_O_3_.

**Figure 3 materials-14-03502-f003:**
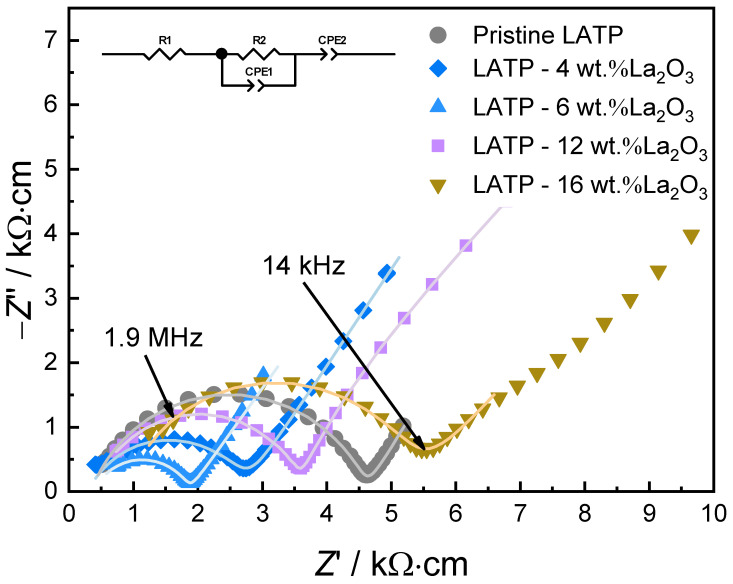
Nyquist plots of pristine LATP and composite samples with fitted curves. The related equivalent circuit is shown in the inset.

**Figure 4 materials-14-03502-f004:**
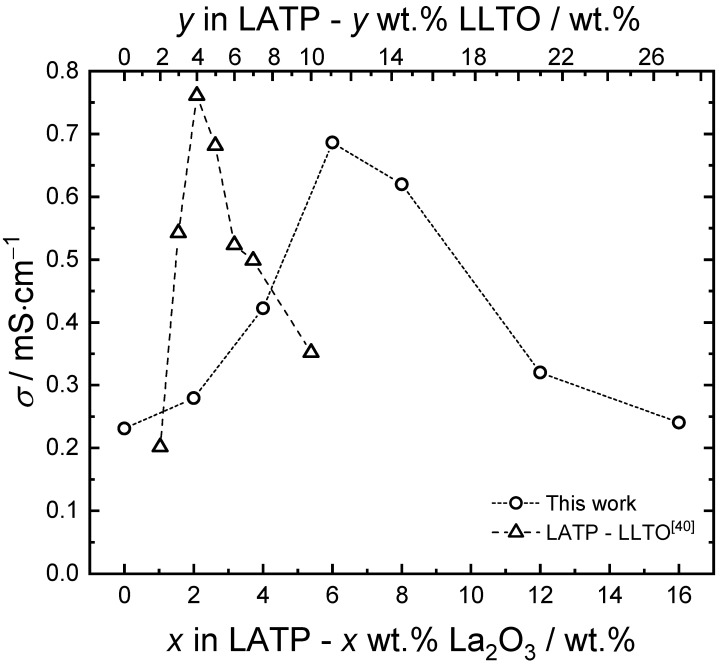
Room temperature conductivity of LATP–x wt.% La_2_O_3_ as a function of La_2_O_3_ addition, in comparison with the results in LATP–y wt.% LLTO from the previous work [40].

**Figure 5 materials-14-03502-f005:**
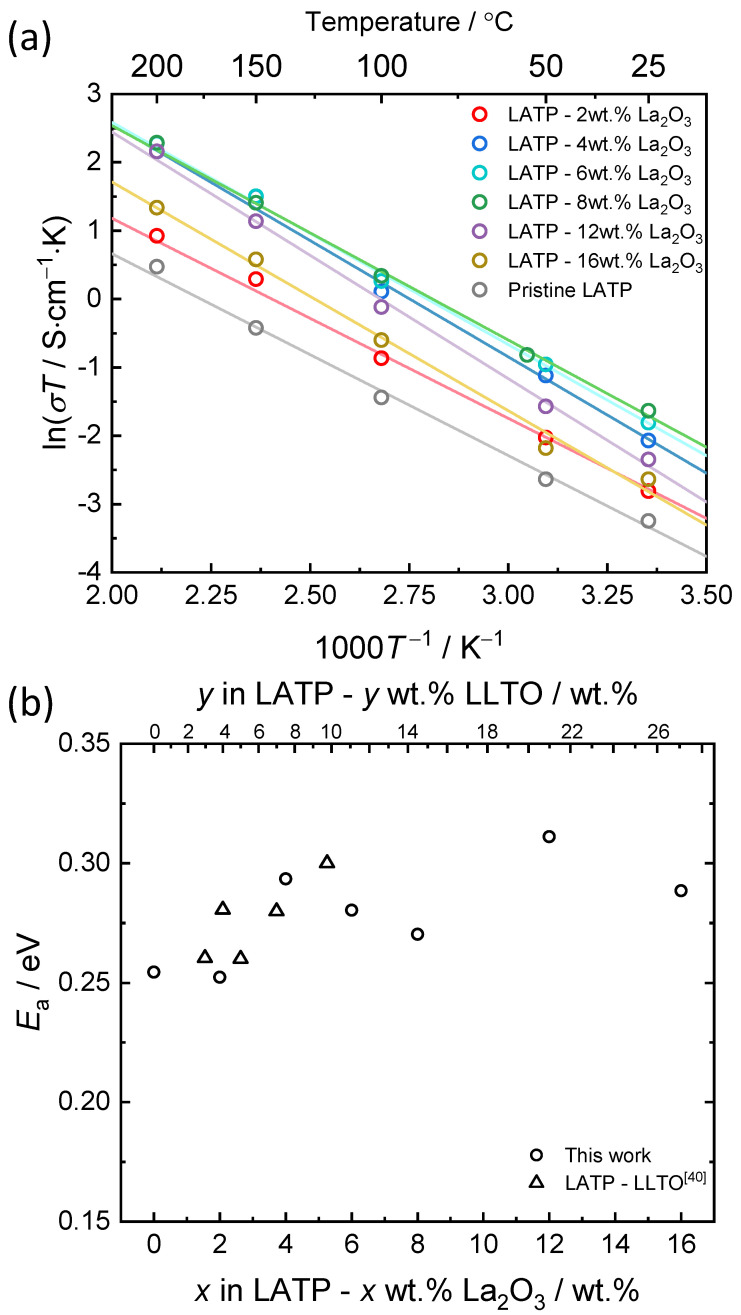
(**a**) Arrhenius plots of LATP–x wt.% La_2_O_3_, and (**b**) activation energies of LATP–x wt.% La_2_O_3_ compared with the results of the previous work [40].

## Data Availability

The data presented in this study are available on request from the corresponding author.

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
