# Peer review of "Synthesis and Characterization of Lithium-Ion Conductive LATP-LaPO4 Composites Using La2O3 Nano-Powder"

_materials, 2021, doi:10.3390/ma14133502_

Round 1

Reviewer 1 Report

Dear Authors,

The article is written well and the results are well described.

You have some impurity in the composite. and It increases with the amount of La2O3. Is this non-reacted La2O3 in the composite?

Could you put the top view of optical or microscopic images of the samples?

Greetings

Reviewer 2 Report

In the submitted manuscript, the authors focus on the introduction of insulative nano-powder of La2O3. They adopted a strategy proposed by C.C. Liang where insulating particles were dispersed in a Li-ion conductor. However, the reason why nano-La2O3 is introduced is not exactly apparent, significantly how it may affect the properties of the studied materials. The manuscript is definitely worth considering for publication, but there are still some issues that need to be solved before publication.

  • Page 2, line 75 – at the preparation stage, it should be noted how much La2O3 was added.
  • Figure 1 – despite the formation of LaPO4 and other unidentified phases, there are still some new reflexes that were not marked on the figure, i.e., at 2θ angles 17.0°, 21.4°, and 27.5°.
  • Figure 1 – except for AlPO4, LiTiPO5 is the most common compound formed during sintering of the LATP material. The diffraction reflexes for LiTiPO5 may be found around 27° so that it may be your unidentified phase. Please, verify if it is lithium titanium phosphate because the too high content of LiTiPO5 may be the reason for lowering total ionic conductivity (some more information may be found in the articles presented at the end).
  • Page 4, line 112 – What are the values of the bulk conductivity from the fitting? Such information should be included in the manuscript. What does the Rs stand for? Have you tried fitting with the R-CPE loop for the bulk?
  • Page 6 Figure 4 – the x scale on the top goes to 26 wt%, but the maximum concentration of LLTO in the composite is 10 wt%, so the scale on top should be reduced.
  • Page 7 Figure 5 – the authors should put error bars on the LATP-LLTO composite or delete the ones from LATP-La2O3 to make the graph uniform.
  • There is a totally lack of discussion with other works concerning the LATP-based composites. Please, refer to the following works:

Kwatek et al. Journal of Alloys and Compounds 838 (2020), 155623.

Kwatek et al. Journal of the European Ceramic Society 40 (2020), 85-93.

Hupfer et al. Solid State Ionics. 302 (2017), 49–53.

Round 2

Reviewer 2 Report

The answers to the comments and corrections introduced in the manuscript are satisfactory. I accept the manuscrpit in the current form.